# Genome maps across 26 human populations reveal population-specific patterns of structural variation

Michal Levy-Sakin[1], Steven Pastor[2], Yulia Mostovoy[1], Le Li [3], Alden K.Y. Leung[4], Jennifer McCaffrey[2], Eleanor Young[2], Ernest T. Lam[5], Alex R. Hastie[5], Karen H.Y. Wong [1], Claire Y.L. Chung [4], Walfred Ma[1], Justin Sibert[2], Ramakrishnan Rajagopalan[2], Nana Jin[4], Eugene Y.C. Chow[4], Catherine Chu[1], Annie Poon[1], Chin Lin[1], Ahmed Naguib[5], Wei-Ping Wang[5], Han Cao[5], Ting-Fung Chan [4,6], Kevin Y. Yip [3,6], Ming Xiao[2,7] & Pui-Yan Kwok [1,8,9]

Large structural variants (SVs) in the human genome are difficult to detect and study by conventional sequencing technologies. With long-range genome analysis platforms, such as optical mapping, one can identify large SVs (>2 kb) across the genome in one experiment. Analyzing optical genome maps of 154 individuals from the 26 populations sequenced in the 1000 Genomes Project, we find that phylogenetic population patterns of large SVs are similar to those of single nucleotide variations in 86% of the human genome, while ~2% of the genome has high structural complexity. We are able to characterize SVs in many intractable regions of the genome, including segmental duplications and subtelomeric, pericentromeric, and acrocentric areas. In addition, we discover ~60 Mb of non-redundant genome content missing in the reference genome sequence assembly. Our results highlight the need for a comprehensive set of alternate haplotypes from different populations to represent SV patterns in the genome.

[1] Cardiovascular Research Institute, University of California–San Francisco, San Francisco, CA 94143, USA. [2] School of Biomedical Engineering, Drexel University, Philadelphia, PA 19104, USA. [3] Department of Computer Science and Engineering, The Chinese University of Hong Kong, Hong Kong SAR, China. [4] School of Life Sciences and State Key Laboratory of Agrobiotechnology, The Chinese University of Hong Kong, Hong Kong SAR, China. [5] Bionano Genomics, San Diego, CA 92121, USA. [6] Hong Kong Bioinformatics Centre, The Chinese University of Hong Kong, Hong Kong SAR, China. [7] Institute of Molecular Medicine and Infectious Disease in the School of Medicine, Drexel University, Philadelphia, PA 19104, USA. [8] Department of Dermatology, University of California–San Francisco, San Francisco, CA 94143, USA. [9] Institute for Human Genetics, University of California–San Francisco, San Francisco, CA 94143, USA. These authors contributed equally: M. Levy-Sakin, S. Pastor, Y. Mostovoy, L. Li, A.K.Y. Leung, J. McCaffrey. These authors jointly supervised this work: T.-F. Chan, K.Y. Yip, M. Xiao, P.-Y. Kwok. Correspondence and requests for materials should be addressed to P.-Y.K. (email: pui.kwok@ucsf.edu)

Among all classes of sequence variation, large structural variants (SVs, >2 kb) are the least studied because they are difficult to detect comprehensively across the genome[1]. Whereas tens of thousands of individuals have been whole-exome or whole-genome sequenced[2–5], genome-wide SV data of only a small number of individuals have been published. As short-read sequencing dominates human genome analysis, large SVs are mostly invisible to such analyses because of the presence of numerous repetitive elements[6]. The diploid nature of the human genome makes it even more difficult to detect large SVs in the heterozygous state[7]. Long-read sequencing methods and other long-range genome analysis platforms (such as optical mapping, Strand-seq, and Hi-C) provide the means to study this previously inaccessible class of sequence variation[8,9]. We have shown previously that optical mapping based on single-molecule analysis is an efficient and reliable way to detect large SVs[10] and, when combined with long-read DNA sequencing, is useful in de novo assembly of the human genome[11]. In this report, we describe the large SV content of 154 individuals across 26 populations sequenced in the 1000 Genomes Project (1KGP)[3,12], the population patterns of these SVs across populations from around the world, the locations of large insertions and additional copies of large copy number variants (CNVs), and the structure of regions that are difficult to characterize (such as segmental duplications (SDs), subtelomeric areas, pericentromeric areas, and acrocentric chromosome p-arms).

Our results show that the reference genome represents only one haplotype among many in the world population, and has substantial missing content (~60 Mb). We observe that the bulk of the genome contains large SVs with phylogenetic population patterns similar to those of single-nucleotide variations (SNVs). However, ~2% of the genome consists of clusters of very large, complex SVs comprising many distinct haplotypes; these loci have substantial overlap with SDs and subtelomeric and pericentromeric regions. These results confirm previous observations that SDs and parts of the genome with complex structural variations behave differently from the rest of the genome[13]. Furthermore, we are able to extend the genome maps into subtelomeric, pericentromeric, and other regions with large tandem repeats that are not characterized currently in the reference genome sequence. Using linked-read sequence data from the 10x Genomics (10xG) Chromium platform[14] for 13 of the individuals, we have validated many of the large SVs. Taken together, our study shows the utility of genome-wide analysis of large SVs and points to the need for an expanded set of alternate reference haplotypes to capture the diversity in SVs across populations.

## Results

**Genome maps of diverse populations**. We selected 156 samples from 26 different populations collected by the 1KGP[3,12]. From each population, genome maps were constructed for 6 biologically unrelated samples (3 males and 3 females) based on de novo assembly of large single molecules (≥150 kb) fluorescently labeled at Nt.BspQI nicking sites[10]. Mapping data were collected at 79× average coverage with molecule length N50 of 262 kb. Two samples failed at the data acquisition step, resulting in genome maps for 154 samples, each with an average of 3427 contigs at N50 size of 1.16 Mb (Supplementary Data 1).

We determined the portions of the reference genome that could theoretically be mapped with the Nt.BspQI nicking enzyme. One class of unmappable reference regions contain large sequence gaps, providing no in silico nicking site information. We explored some of these regions by analyzing maps aligned to flanking regions (discussed below). Additional long sequences in the genome (e.g. much of the centromere) lack Nt.BspQI motifs,

resulting in featureless DNA molecules. We defined reference regions as inaccessible for mapping where the sequence-based gaps were ≥50 kb (after removing gaps that were fully spanned with mapping data; see Methods) or the stretches lacking Nt. BspQI motifs were ≥100 kb (Fig. 1a, Supplementary Data 2). Using these criteria, 2.87 Gb of the genome (93%) can be mapped, with most inaccessible regions found in centromeric, pericentromeric, and subtelomeric regions. The 154 de novo genome assemblies aligned to an average of 2.53 Gb of hg38, with one sample, NA19239, aligned to 2.85 Gb (covering ≥99% of the mapping-accessible reference genome). Samples that aligned to shorter lengths of hg38 had smaller de novo assemblies associated with shallower molecule coverage depth and/or shorter molecule length (Supplementary Data 1). In addition, assemblies contained maps not aligned to the hg38 reference, totaling ~60 Mb in aggregate over the 154 samples (described below).

**Linked-read genome assembly and hybrid assembly**. For each of the 13 subpopulations deemed most genetically distinct from one another[3], we selected one sample with a high-quality genome map assembly to obtain whole-genome linked-read sequencing data using 10xG Chromium sequencing libraries. These were sequenced to ~60× average coverage, with a mean inferred molecule length of 82 kb. Phased genome assemblies were generated using Supernova software (v1.1)[15], yielding an average of ~1300 scaffolds (of >10 kb) totaling ~2.7 Gb, with N50s ranging between 14.7 and 23.5 Mb for each assembly[16].

In addition, we generated hybrid scaffolds where genome maps were used to bridge the Supernova scaffolds. The hybrid assemblies contained ~2.8 Gb in 188–265 scaffolds at N50s of 25–35 Mb, with some of the scaffolds covering full chromosomal arms (see Supplementary Fig. 1 and Supplementary Table 1). Edges of scaffolds were typically near SDs.

**Classification of structurally complex regions**. While SVs are found throughout the genome, we sought to identify loci that harbored multiple structurally distinct haplotypes with clusters of large SVs (Supplementary Fig. 2). To guide this search, we assembled the genome maps from all 154 samples into a single consensus assembly that consisted of 1245 maps spanning 2.85 Gb, covering 99.3% of the mapping-accessible reference genome and capturing the predominant SVs in the dataset (Supplementary Fig. 3). The vast majority of the accessible genome was well-covered by individual genome maps and assembled into one consensus scaffold per locus, reflecting low levels of structural diversity (Fig. 1b, c). In contrast, 83 Mb were covered either by multiple overlapping consensus scaffolds, or by no consensus scaffolds despite being well-covered by individual genome maps, reflecting loci too structurally diverse to represent with a single consensus scaffold. These regions were manually curated using our criteria of structural complexity: containing ≥3 different SVs within the full dataset, including inversions, translocations, CNVs, and insertions/deletions (indels) >10 kb. For loci containing only indels, at least one of the indels had to be >100 kb. Borders of each complex area were manually adjusted to exclude structurally simple flanking areas. After this curation, 55 loci (67 Mb) were defined as structurally complex (Fig. 1a, d; Supplementary Data 3), and 2.65 Gb was classified as having low structural complexity. Features known to be predisposed to structural complexity–subtelomeric and pericentromeric regions, SDs, and long tandem repeats–were each overrepresented in the complex regions compared to the entire accessible reference genome (resampling $p \leq 5e-5$). Many of the complex regions were associated with pathogenic variation in the ClinVar[17] and OMIM databases[18] (Supplementary Data 3).

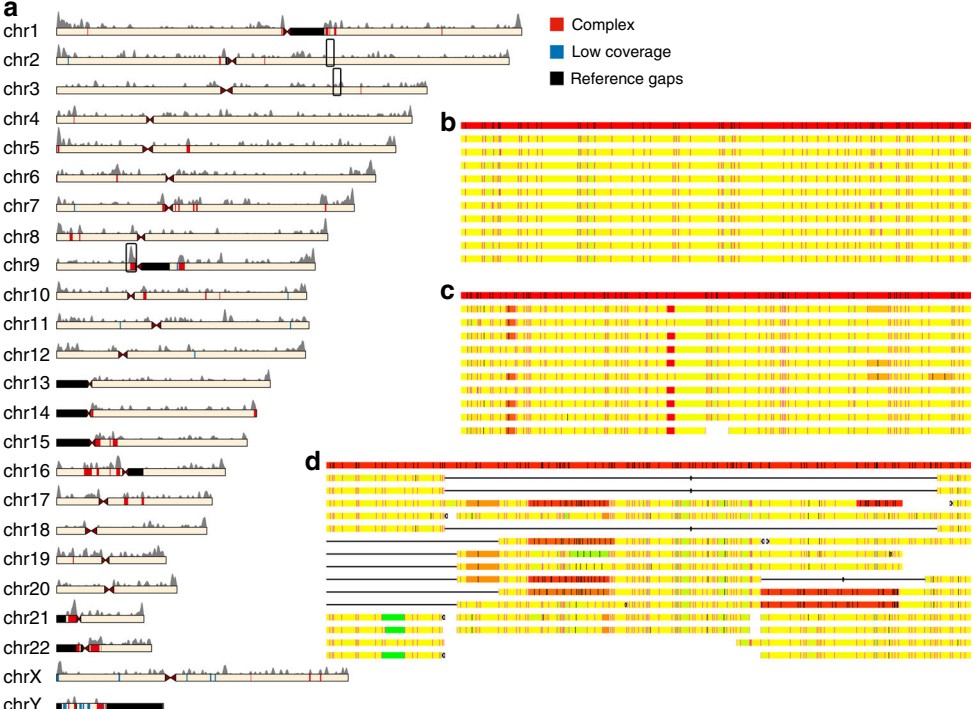

**Fig. 1** Distribution and properties of large structural variations in the human genome. **a** Ideogram depicting the distribution of indels and complex regions. The gray histogram above the chromosomes depicts the number of indels detected in all populations using a sliding window of 1 Mb with a 10-kb step size. Chromosome fill shows the different regions classified in the genome. Red, structurally complex regions; blue, low individual assembly coverage; black, regions with long sequence- or nick-based gaps in the reference. For display purposes, both low-coverage and gap regions were only displayed if they were longer than 500 kb. Black boxes enclose areas shown in more detail in **b**–**d**. **b** Alignment of individual sample assemblies to the reference at chr2: 148.4–149 Mb. This region contains a low level of structural variation on the basis of both indel calls and the consensus assembly. **c** Alignment of individual sample assemblies to the reference at chr3: 38.4–39 Mb. This region contains a high density of indel calls but a low level of complex structural variation, and was categorized as low complexity. **d** Alignment of individual sample assemblies to the reference at chr9: 42–42.9 Mb. This pericentromeric region has complex structural variation as well as many indels. In **b**–**d**, the red bar at top is the reference nick pattern and the bars below are consensus maps from 10 random samples. Yellow segments are highly similar to the reference, while green or red segments are shorter or longer than the corresponding reference segments, respectively, where intensity of color correlates positively with difference from the reference. Red lines represent aligned labels and black lines represent unaligned labels. Circles denote inversions while squares at the ends of contigs denote translocation break ends

**Structural variation identification and validation**. We used a modified version of OMSV[19] to identify large (>2 kb) SVs from each of the 154 samples (Supplementary Data 4–8; Supplementary Fig. 4). In total, we identified 15,601 unique high-confidence large indel SVs (average size 11.2 kb, largest insertion at 381 kb, Table 1). Each sample has a median of 1539 large indels that span 14.2 Mb. Among the large indels, 14,226 (91%) are in low-complexity regions. Interestingly, while large indels generally display a smooth size distribution, there is a clear peak ~6 kb (Supplementary Fig. 5), corresponding to LINE1 elements. In addition, we also detected 934 complex SVs, including inversions, SDs, and loci with multiple large indels in close proximity (Table 2; Supplementary Fig. 6).

We used three independent methods to evaluate the accuracy of the large indels detected. First, we checked whether they could also be found in the 10xG linked-read sequencing data from 13 samples (Methods; Supplementary Fig. 7). Among the ones that could be examined, 78% (90% of the deletions and 69% of the insertions) were confirmed by the 10xG contigs (Supplementary Fig. 8). Further checking against the calls in Sudmant et al.[12], 84% of our large indel calls (95% of deletions and 76% of insertions) were confirmed by either the 10xG data or the Sudmant study[12] (see Supplementary Table 2).

Second, we generated genome map data from additional family members for four samples, resulting in data for 4 trios

(Supplementary Table 3). We applied our pipeline to identify large indels in these 12 samples independently and found that 95.4% of them were concordant with Mendelian inheritance (Supplementary Table 3).

Third, we found that among the large indels identified from sex chromosomes of female samples, only 0.1% were wrongly called from the Y chromosome.

Taken together, these results confirm the high accuracy of the large indels detected.

Among the samples in this study, 144 were also included in a recent study by the 1000 Genomes Project[12]. For the 15,024 large indels we identified from these samples, 3966 (26.4%) overlapped those identified in that study (Fig. 2a). Most of the remaining novel large indels were insertions (Fig. 2b, c), consistent with previous reports that large insertions are difficult to detect by short-read sequencing[1,19,20]. Importantly, many of these novel insertions were independently detected in multiple samples, and some of the inserted sequences were contained in alternative loci in the hg38 reference (Supplementary Table 2), both supporting the existence of these novel insertions. In all, 1647 (25%) of the large indels identified in the 1KGP[12] did not overlap our calls. When we reduced the stringency of OMSV, 1252 of them were recovered. We further investigated the remaining 395 calls and found that approximately one-third of them (123) were in inaccessible regions due to the nicking enzyme used for genome

**Table 1 Number of large (>2 kb) insertions and deletions identified from the 154 samples**

| | Type | Total number of unique SVs | Average size (bp) | Maximum size (bp) | Median number per individual | Median SV allele per individual | Median size (kb) per individual |
|---|---|---|---|---|---|---|---|
| High-confidence list | Insertions | 6495 | 14,082 | 380,886 | 906 | 1513 | 6611 |
| | Deletions | 9106 | 9160 | 259,478 | 633 | 1044 | 7568 |
| | Total loci | 15,601 | 11,209 | 380,886 | 1539 | 2557 | 14,179 |
| Filtered high-confidence list with complex regions removed | Insertions | 5839 | 12,429 | 288,377 | 816 | 1367 | 5001 |
| | Deletions | 8387 | 8036 | 225,078 | 524 | 848 | 3788 |
| | Total loci | 14,226 | 9839 | 288,377 | 1340 | 2224 | 8893 |

*SV structural variant*

**Table 2 Number of complex structural variations identified from the 154 samples**

| Type | Total number of unique SVs |
|---|---|
| Inversions | 380 |
| Duplications | 47 |
| Intra-chromosomal break ends | 11 |
| Inter-chromosomal break ends | 25 |
| Multiple insertions/deletions | 458 |
| Others | 13 |

*SV structural variant*

mapping. For the remaining 272 large indels (4% of the 6526 SVs identified by the 1KGP[12] from these 144 samples), the majority of our genome maps aligned to these loci supported the reference allele only.

We further compared our large deletions with those identified in several large-scale studies[12,21,22]. Among the 8700 large deletions we identified, 5762 were also found in these studies (Fig. 2d), suggesting that the remaining 34% of our SVs were novel. Of the 718 large deletions and 877 large insertions we identified in the widely studied NA12878 sample, 562 deletions and 383 insertions were also found in the previous studies with SV data[12,23–26], suggesting that 22% of our deletions and 44% of our insertions were novel.

The novel deletions we identified were mostly rare variants (found only in one sample, 63%), or found in samples not included in the other studies, or missed by previous studies due to low sequencing depth[12]. With 90% of our deletions called in one sample supported by 10xG data in general, many of these novel deletion calls are likely correct.

**Population structure of large insertions/deletions**. We explored the population structure of large indels in the low-complexity regions at three levels. At the super-population level, after eliminating effects due to sample number differences (Methods), we investigated the specificity of the large indels (Fig. 3). In general, 30–44% of the large indels identified from each super-population are shared by all five super-populations, 26–36% are shared by some but not all super-populations, and 22–44% are unique to one super-population, with the largest fraction of the uniquely identified SVs found in Africans followed by East Asians.

At the population level, we performed a phylogenetic analysis based on the occurrence of large indels in their samples of each population (Fig. 3b). Populations from each super-population largely clustered together, and the African cluster was first separated from the other four super-populations. A similar

grouping of samples was also observed from a clustering analysis of the populations (Supplementary Fig. 9).

At the individual sample level, we performed a principal component analysis (PCA) of the SV allele counts (Fig. 3, Supplementary Fig. 10). We found that the first PC almost completely separated African samples from the other four super-populations, while the second PC clearly separated European, South Asian, and East Asian samples, with American samples admixed with the European and South Asian clusters.

We also compared the SV sizes in different super-populations. An analysis of variance led to 85 large indels with statistically significant size differences among super-populations (called by at least 10 samples, significance level of 0.05 with Bonferroni correction; see Supplementary Data 9 and Supplementary Fig. 11).

When these analyses were repeated for insertions and deletions separately (Supplementary Fig. 12), the resulting trends remained highly similar, with the only notable difference being a larger proportion of super-population-specific deletions than insertions.

**Population pattern of copy number variations**. Complex CNVs with large repeat units and numerous copies have been challenging in genome analysis. Genome maps built from long molecules can span large tandem repeat units and present an advantage in characterizing such CNVs[10,19]. Using a multiple alignment approach (see Methods), we performed quantitative analysis of CNVs by aligning contigs of these regions across all 154 samples (Supplementary Fig. 13). Once the consensus flanking regions of a CNV are defined, it is possible to determine the exact copy number in each sample. The results are summarized in Supplementary Data 10. When analyzed across the population groups, patterns were observed at several loci such as the pepsinogen locus at chromosome 11q12.2 (Fig. 4). An average of 2.5, 2.1, 3.3, 1.6, and 2.4 copies of a 20-kb repeat unit were observed in the Africans (AFR), Americans (AMR), East Asians (EAS), Europeans (EUR), and South Asians (SAS) super-populations, respectively, with a significantly higher copy number ($p < 0.05$, Tukey test) in the EAS super-population. We compared the list of CNVs with functional categories including RefSeq gene annotation[27], SDs[28], the ClinGen CNV database[28], and the NHGRI-EBI GWAS Catalog[29] (Supplementary Table 4). We found that all CNVs were associated with at least one of these features, suggesting that these regions are hotspots for variation and some could lead to the generation of pathogenic variants.

**Characterization of complex regions**. Our analysis of the consensus and individual genome assemblies identified 55 large complex regions, 27 of which were located around centromeres and telomeres, regions that were poorly assembled in the human reference genome. Genome maps built with long molecules from multiple individuals can characterize many of these complex

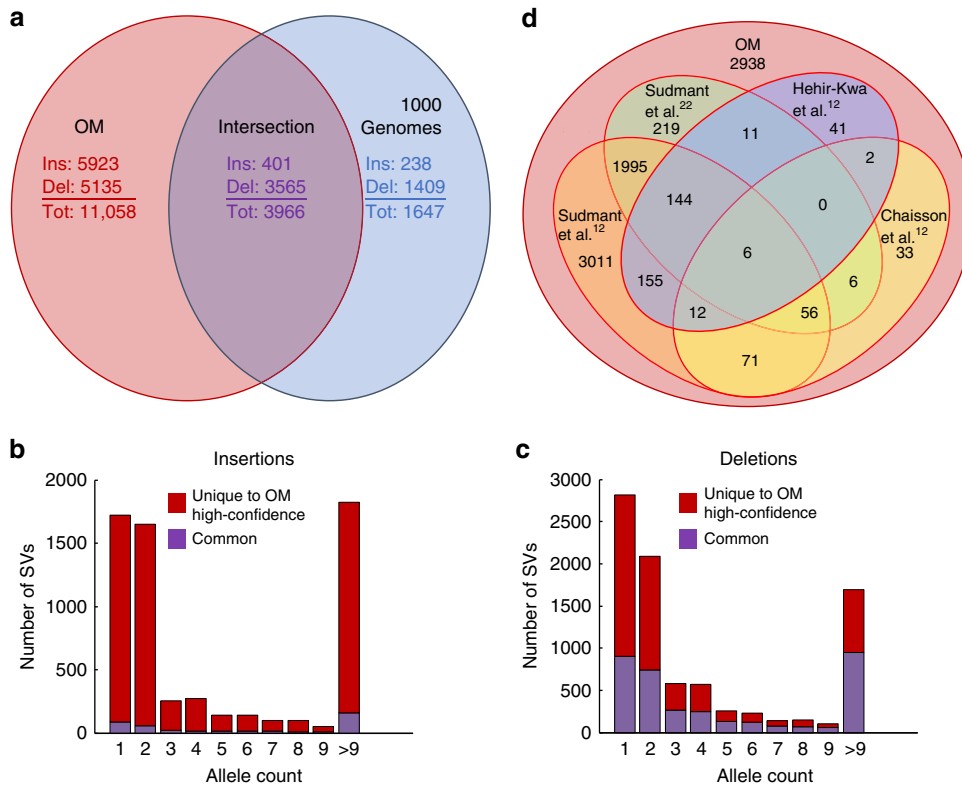

**Fig. 2** Comparisons between the large indels identified by optical mapping (OM) in this study and the ones identified in other large-scale studies. **a** Number of overlapping and unique large indels identified in our study and Sudmant et al.[12] based on the 144 samples commonly studied in the two studies (1KGP). Ins and Del correspond to insertions and deletions, respectively. Tot is the total number of indels in each category. Numbers in red, blue, and purple correspond to numbers of indels identified by OM only, by 1KGP only, and by both, respectively. Since one OMSV may overlap multiple 1KGP SVs, and vice versa, there are actually two sets of numbers in the intersection between the OM set and the 1KGP set. To keep the Venn diagram simple, we have only shown the numbers of indels in the OM set that overlap indels in the 1KGP set. This intersection also contains 406 insertions and 4473 deletions (4879 indels in total) in the 1KGP set that overlap indels in the OM set. **b, c** Distributions of allele frequencies of the indels uniquely identified by optical mapping (red) and commonly identified in this study and Sudmant et al.[12] based on the 144 common samples (purple), considering only insertions (**b**) and only deletions (**c**). Since each sample has at most two SV alleles, for an SV with an allele count of $x$, the number of samples that support this SV is between $\lceil x \rceil$ and $x$. **d** Number of large deletions identified by optical mapping that are also identified in any samples in four other studies (see also Supplementary Fig. 27)

regions and fill in the gaps found in the reference genome. As an example, we characterized a 1.3-Mb complex region near the centromere of chromosome 21 (21p11.2; chr21: 9,500,000–10,800,000). We identified key differences in genome structure between hg38 and our proposed structure informed by multiple alignment analysis of 154 samples (Supplementary Fig. 14). Figure 5 shows the region in hg38 starting from subregion C, followed by F2 and ending up with E (indicated by blue arrows), separated by large unfilled gaps. Multiple alignment analysis on the contigs aligned to these subregions suggested a different genomic structure with a new order of subregions (Fig. 5a, indicated by red arrows). A comparison of the genome structure between hg38 and the one based on multiple alignment is shown in Fig. 5b. Several subregions (A, B1, and B2) not present in chr21 of hg38 were identified, with population-associated haplotypes across the region. For example, the AMR and EUR super-populations had much lower occurrence of D2 comparing to the other populations (Supplementary Fig. 15).

**Characterization of SDs in subtelomeric regions**. SDs contribute to human evolution, adaptation, and genomic instability but are poorly characterized[30,31]. SDs represent ~5% of the genome and are frequently found near centromeres and telomeres[32–34]. Among 126 Mb of known SDs (≥10 kb in length and

≥95% sequence identity), 49 Mb were in the complex regions, 36 Mb were in inaccessible or low-coverage regions, and the remaining 41 Mb were scattered across 2.65 Gb of low structural complexity regions. On average, SDs in complex regions (median 82 kb) were significantly longer than SDs in low structural complexity regions (median 32 kb) and had significantly higher sequence identity (median 99.4% vs 97.4%). These long and extremely similar SDs are intractable with most sequencing technologies but can be reconstructed using long optical map molecules.

Human subtelomeric regions contain inter-chromosomal SDs[35], including paralogy block 3, a 40-kb segment of DNA found in subtelomeric regions on multiple human chromosome arms. However, the hg38 reference does not extend to the block 3 region. The genome maps from long molecules consistently extended beyond the ends of chromosomes in hg38, and detected this block on 13 chromosome arms. Figure 6 shows the paralogous blocks of subtelomeres combined differently on each chromosome arm, and between haplotypes. Some contain blocks missing from the hg38 reference or Stong assembly[36]. The distribution of block 3 on 6p was highly variable. For 15q, the majority of haplotypes contained block 3. For 7p, most of the samples had extra blocks that were not present in the hg38 reference. HG00353 had blocks 1–9 (185 kb), while NA18986 contained an extra 90 kb, but lacked block 3. Chromosome arms

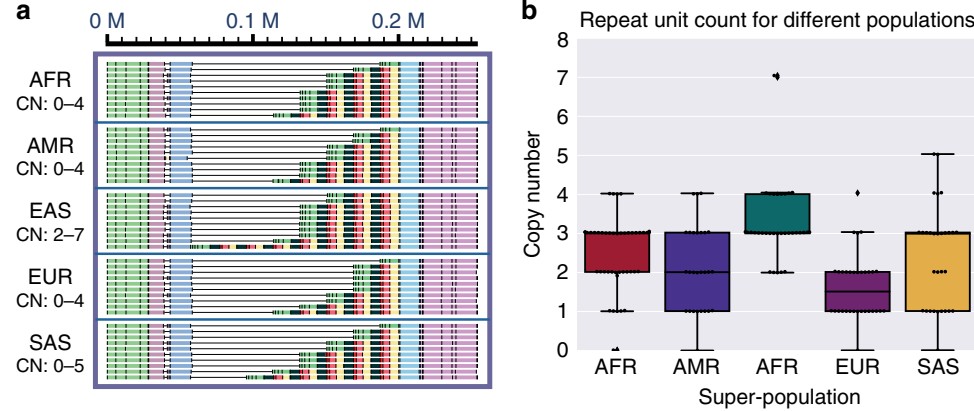

**Fig. 3** Population structure of large indels at three different levels. **a** Super-population level: the average ratio of indels identified from samples in each super-population that are specific to that super-population, shared with some other super-populations but not all, or shared with all other super-populations. Random sub-sampling has been applied to balance the sizes of super-populations. The reported values are the average of 100 random sub-samples. **b** Population level: a phylogenetic tree constructed based on the indel occurrence matrix. **c** Single-sample level: the first two principal components of the indel occurrence matrix based on super-population groups. AFR Africans, AMR Americans, EAS East Asians, EUR Europeans, SAS South Asians

**Fig. 4** Analysis of copy number variant (CNV) in pepsinogen A cluster at chromosome 11q12.2 using multiple alignment. **a** Visualization of CNV haplotypes using multiple alignment of contigs allows easy counting of the repeating units. Copy numbers (CNs) are shown on the left. Alignment of all individuals across the 26 populations is provided in Supplementary Fig. 13. **b** A boxplot for CN in different ethnic groups. East Asians (EAS) have significantly more copies than other populations ($p < 0.05$, Tukey test), while Europeans (EUR) have significantly fewer copies than other populations ($p < 0.05$, Tukey test) except Americans (AMR)

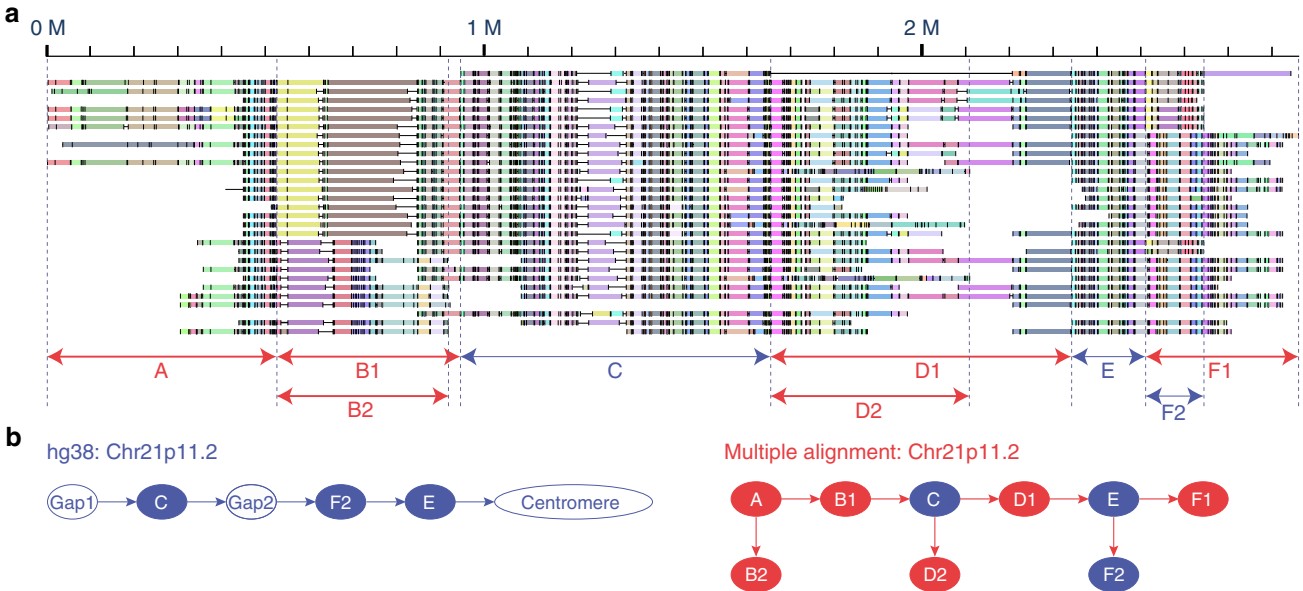

**Fig. 5** Characterization and extension of chromosome 21p11.2 by multiple alignment. **a** Multiple alignment of selected contigs at 21p11.2. Red and blue arrows below indicate the subregions with patterns absent and present in chr21 of hg38, respectively. **b** The genome structures of hg38 (left) and the proposed new structure based on multiple alignment (right). The genome structure is represented as a flow chart of signal patterns where red and blue nodes represent patterns absent and present in chr21 of hg38, respectively

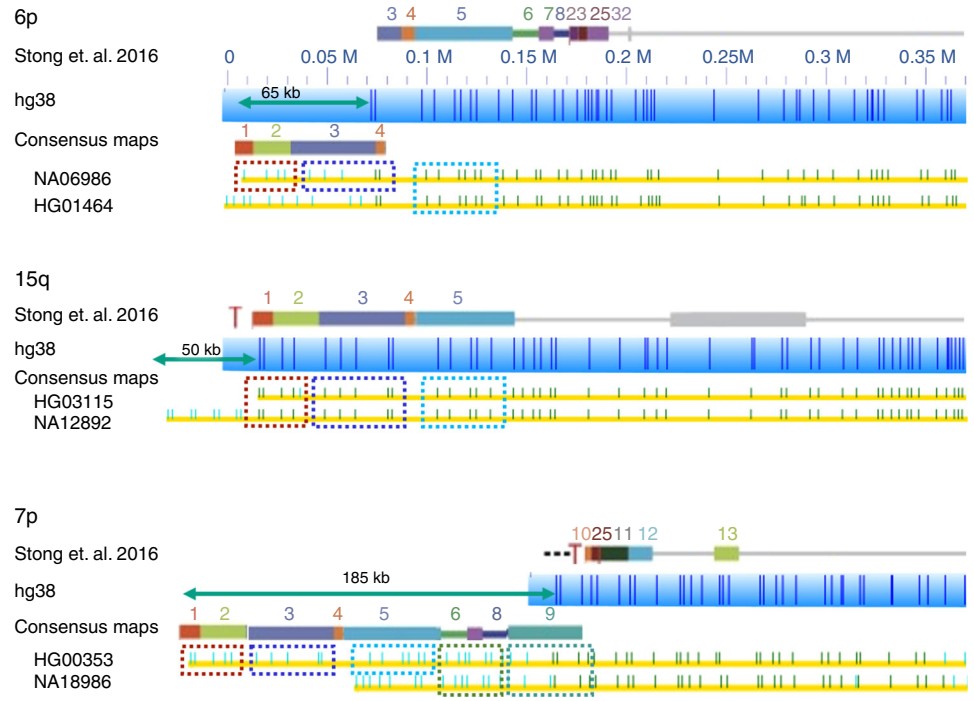

**Fig. 6** Chromosome- and population-dependent distribution of paralogy block 3 in subtelomeric regions. Previous subtelomere repeat elements paralogy block references, shown as colored rectangles above each chromosome arm[36]. Yellow rows depict consensus contigs beneath blue bars representing hg38 references. Dark green dashes indicate Nt.BspQI nick sites matching the reference while lighter green dashes represent unmatched nick sites. Teal arrows measure regions not in either reference. Additional paralogy blocks are also shown above these extended regions with dashed boxes indicating regions matching specific blocks. Paralogy block 3, shown by the dashed box in purple, is found on only one haplotype of 6p, on all haplotypes of 15q, and on a less common haplotype of 7p

3q, 6q, 15q, and 19p showed a heavy prevalence of block 3 in >50% of the individuals, with similar frequency among super-populations. Seven percent had block 3 in the 7p arm, and 64% of those were from the African super-population. The distribution in 16p and 16q arms behaved similarly, with 3 and 14% of all individuals had block 3 in their 16p and 16q arms. Among those individuals with block 3, 75% (16p) and 71% (16q) were from the African super-population. Block 3 occurrence and patterns were

chromosome- and population-dependent. Our results support the findings of a previous study of this paralogy block that signified recent human divergence[37].

**Novel genome content not in the reference**. We found 4063 consensus maps across 154 samples that could not be aligned to the hg38 reference. In aggregate, ~60 Mb of new genome content was identified. To find the unique non-aligned consensus maps, we used an all-against-all comparison and generated 53 unique groups from 2224 non-aligned consensus maps. The remaining 1839 consensus maps were categorized as tandem repeats.

To determine if the ~60 Mb of new content is found in other published genome assemblies, we constructed in silico nick-labeled sequence maps from several genome sequence assemblies for comparison. We examined the hg38 Alternative, Random, and Uncharacterized (decoy) sequences[38], the Chinese genome assembly (HX1)[39], the Korean genome assembly (AK1)[11], and 11 primate reference genomes (*Callithrix jacchus* (GCA_000004665.1), *Chlorocebus sabaeus* (GCA_000409795.2), *Gorilla gorilla* (GCA_000151905.3), *Macaca fascicularis* (GCA_000364345.1), *Macaca mulatta* (GCA_000772875.3), *Nasalis larvatus* (GCA_000772465.1), *Nomascus leucogenys* (GCA_000146795.3), *Pan paniscus* (GCA_000258655.2), *Pan troglodytes* (GCA_002880755.3), *Papio anubis* (GCA_000264685.2), and *Pongo abelii* (GCA_002880775.3)) and found that ~36.5 Mb of our non-aligning content are not found in these assemblies. Specifically, 14 of the 16 HX1 maps not aligned to the reference were found in our content. Overall, 6.8 Mb of the new content aligned to the HX1 genome. For the AK1, 4.6 Mb of its non-aligned content matches our content. We also found that 2.1 Mb of the new content could be mapped to at least one of the 11 primate genomes (see Supplementary Fig. 16 and Supplementary Table 5).

Comparing the non-aligned content of the probands of two trios (Han Chinese-HG00514 and CEPH-NA12878) to their respective parents, we found that 15/20 and 15/17 non-aligned maps from the Chinese and CEPH probands, respectively, aligned to parental maps.

The newly identified maps of the p-arm of the acrocentric chromosomes are part of this content (see below). There is no evidence that the non-aligned content is from viral or bacterial contamination (and we accounted for the Epstein-Barr viral genome in the immortalized cell lines we used in this study).

**Identification of unique acrocentric chromosome patterns**. Non-aligned maps were analyzed for specific patterns, and it was found that the most abundant non-aligned contigs could belong to the p-arms of acrocentric chromosomes. Figure 7 illustrates the most abundant, unique non-aligned maps. These maps shared a 40-kb region with a unique nicking pattern (Fig. 7a, red bar). This segment was flanked by variable counts of ~6 kb tandem repeat units (50–220 kb). The 40-kb segment nicking pattern was identical to one found on the chromosome 4p arm in hg38, and corresponded to an SD shared by subtelomeres of chromosome 4p and 4q. Youngman et al. sequenced 27-kb homologs (blue teeth in Fig. 7a) from acrocentric subtelomeres, which also shared this distinct pattern[40]. Further evidence for the localization of these maps to the ends of chromosomes was obtained through a CRISPR-Cas9 labeling technique, which specifically tagged the telomeric repeats[41]. CRISPR-Cas9-labeled single DNA molecules (Fig. 7a) indicated that telomeric repeats, denoted by the bright green ends, were immediately upstream of the 40-kb segment.

Using the aforementioned features, seven unique acrocentric patterns were discovered (Fig. 7b). These patterns were observed consistently in non-aligned consensus maps across all genomes and were the most abundant patterns found in non-aligned maps. Aside from the 40-kb segment and tandem repeats, these maps contained three additional unifying features, shown in Fig. 7c, for a total of five features we used to define an acrocentric map. Each contained the required 40-kb segment, two sets of tandem repeats, and two separate label distribution groups, which may vary between molecules within a genome. The primary variation of these maps between and within genomes were the copy numbers of the first tandem repeat pattern (TR1). Whole-genome molecule alignments provided evidence of the existence of at least one of the seven acrocentric maps in all genomes with any variation observed in the copy number of labels in TR2. Currently, we cannot localize the acrocentric patterns to specific chromosome arms.

**Characterization of reference N-gaps**. The reference genome contains 603 N-gaps predicted to span 151 Mb, much of which is found in heterochromatin regions (93% by length). To determine which gaps could be better characterized with genome map data, the assembly-to-reference alignment for each sample was evaluated to identify reference gaps that were spanned by contigs. Ninety-two such gaps (3.1 Mb by reference length) were identified, and their map-based gap size was determined (Supplementary Table 6). Most of these gaps (90% by reference length) were found in euchromatin, closing 28% of the euchromatin predicted gap length. The map-based gap lengths were correlated to the reference gap lengths (Pearson's correlation $r = 0.56$, $p = 9e-9$) (Supplementary Fig. 17). Map-based gap lengths ranged from −182 kb, representing a deletion including the gap's flanking sequence, to 428 kb, with a median length of 977 bp.

A complementary analysis was performed using 10xG sequencing data from 13 samples, yielding a total of 113 distinct closed reference gaps with a median length of 375 bp (Supplementary Data 11). As with the genome map data, most of the closed gap length belonged to euchromatin (66% by reference length). Among the assembled gap sequence content, 67% was composed of Ns, likely due to the repetitive nature of gap-containing regions that make them challenging to assemble. Nonetheless, 85 of the assembled gaps contained non-N sequence in at least one sample. Using RepeatMasker[42] on assembled gap sequences, annotations were applied to 66% of the non-N sequence, with the largest proportion (34% by length) annotated as satellite DNA, consistent with the centromeric and pericentromeric locations of many gaps. Other frequent annotations included tandem repeats (24%), LINE1 elements (20%), and long terminal repeats (10%). To gauge the accuracy of assembled gap sizes, we compared the lengths of gaps spanned with both 10xG and genome map data (Supplementary Fig. 18). The two datasets closed 39 of the same gaps, which showed a strong correlation between the genome map and 10xG lengths (Pearson's correlation $r = 0.76$, $p = 1.5e-8$).

**Genome distribution of variable number tandem repeat/minisatellite sequences**. It is likely that many of the large SVs in the genome (especially those 2–20 kb in size) are associated with variable number tandem repeats. We examined the 13 samples for which we had 10xG data and found that almost a quarter of the 4922 SVs identified by optical mapping (23.9% of the insertions and 23.5% of the deletions) are within 20 kb of SDs, telomeres, centromeres, or gaps, rendering the 10xG data not useful for comparison because the SV calls in these regions are unreliable[43].

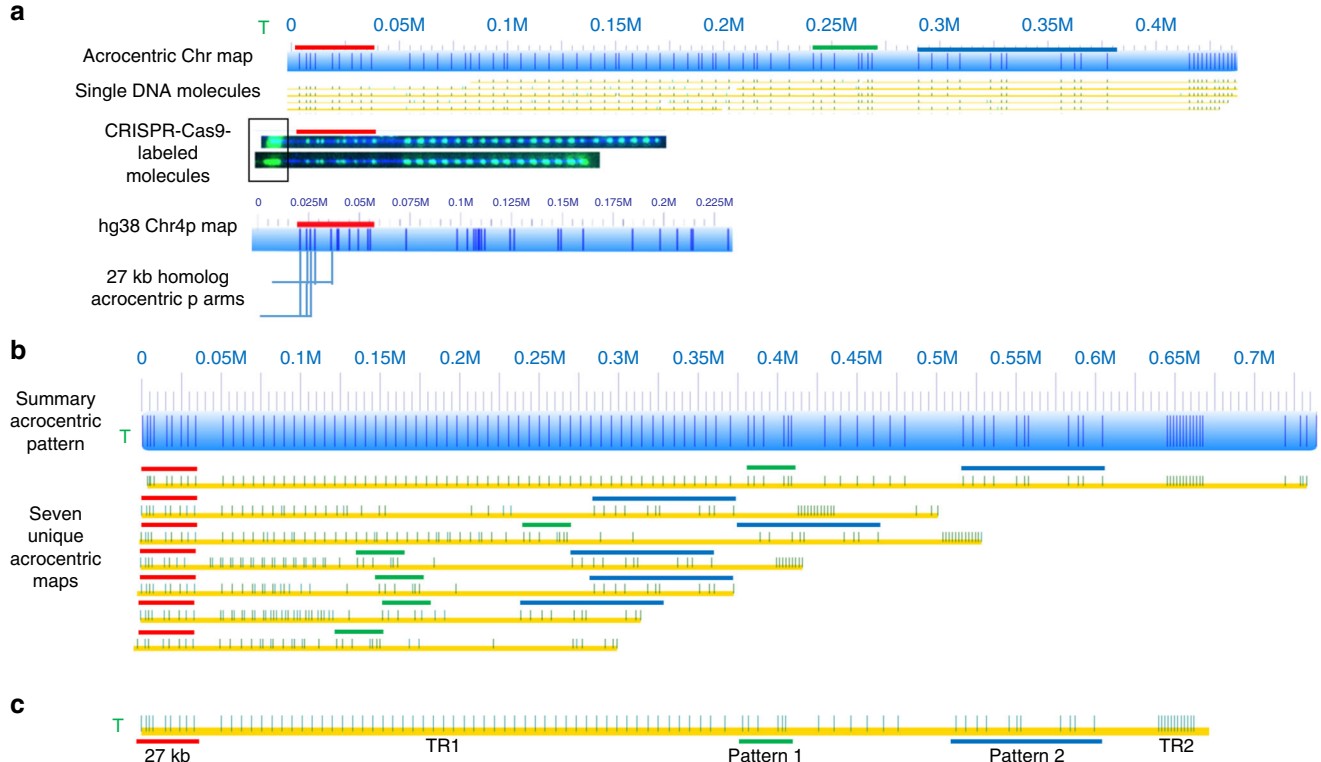

**Fig. 7** Acrocentric chromosome patterns in non-aligned maps. Non-aligned maps were grouped into unique patterns. These groups were analyzed for patterns localizing maps to near telomeric regions. **a** An acrocentric map (blue) supported by molecules (yellow) with Nt.BspQI labels (green dots) aligned to an in silico labeled map (dark blue vertical lines). Molecules comprising the maps were localized to acrocentric regions by a CRISPR-Cas9 labeling method described previously. The green T indicates telomere labeling. The 27-kb homolog, shown by the blue teeth underneath the chromosome 4p map, was identified by in silico nicking of the chromosome 4p sequence indicated by red bars throughout the figure. **b** Seven unique acrocentric maps (yellow bars with green BspQI labels) were identified and aligned to a summarized version (blue bar). The red-, green-, and blue-colored bars indicate the elements defining an acrocentric map. **c** A general model for an expected acrocentric map. From the telomeric end (green T), the 27-kb homolog must be present (red bar), followed by tandem repeat region 1 (TR1), a unique labeling pattern (green bar, Pattern 1), a second unique labeling pattern (blue bar, Pattern 2), and a final tandem repeat region (TR2). Patterns 1 and 2 exhibit little variation, while unit counts of the tandem repeat regions varied considerably

For the remaining SVs (1863 insertions and 1893 deletions), 76.8% and 89.0% of the insertions and deletions, respectively, are supported by 10xG sequencing data. Based on the breakpoints identified by the sequencing data, we used RepeatMasker to classify the SVs and their flanking sequences (see Methods and Supplementary Figs 19–21). We found that interspersed sequences were highly enriched in proximity to SVs, with 87.1% of the insertions and 96.4% of the deletions within 500 bp of repeats/transposable elements (TE) at one or both breakpoints. More than half of the deletions were within 500 bp of LINEs and about 40% of the SVs were within 500 bp SINEs (Supplementary Fig. 19). In the majority of the SVs, TE were found at both ends of the SV, with LINE/LINE the most common combination for deletions and SINE/SINE the most common for insertions (Supplementary Fig. 21). Combinations of mobile elements from two different families such as LINE/SINE were also observed. In addition, we found that a small subset of the SVs could be linked to retro transposition activity, resulting in integration (or deletion) of processed mRNA in the form of pseudogenes (see Methods and Supplementary Figs 19–21).

**Y chromosome analysis**. We generated Y chromosome contigs from 77 male samples that covered the highly duplicated regions of the Y chromosome (Supplementary Fig. 22) and bridged across the many discontinuous segments of the complex repeated structure. Samples with higher coverage depth had a more contiguous

assembly ($p = 1.05e\text{-}15$; Pearson's correlation $r = 0.76$) than those with lower depth, with a median assembly size of 11.2 Mb across all samples. When we analyzed the samples with >100× genome-wide molecule coverage (26% of the sample cohort), the median size of the Y chromosome assembly increased to 18.2 Mb. We identified a total of three deletions, two insertions, two inversions, and two CNVs (Supplementary Table 7) in these samples. Two out of the three deletions corresponded to two large gaps in the primary hg38 assembly and the corrected sequences were found in the fix patches recently submitted (Scaffold ID: KZ208923.1 and KZ208924.1). All other SVs showed a wide range of allele frequencies both within and between population groups (Supplementary Fig. 23 and Supplementary Table 8). In addition, we detected two CNVs in regions that contained four genes in the *RBMY1* family. The first locus harbors *RBMY1A1* and *RBMY1B* while the second one harbors *RBMY1D* and *RBMY1E*. The mean population copy number for the loci were 4 and 2, respectively. This combination of copy number (4 and 2) was the most prevalent in our dataset, representing 45% of the total samples we analyzed (Supplementary Fig. 24). While technology such as array comparative genomic hybridization (aCGH) can provide the overall sum of *RBMY1* copy number, aCGH cannot distinguish the count and the orientation of the specific genes. Overall, these results suggest that the optical mapping strategy can be used to ascertain SVs even in the complex repeated structure of the Y chromosome. However, optical mapping (and other long-read

sequencing platforms) requires unique anchors to place the reads/ molecules in the proper orientation in order to characterize near-identical repeats. Because of this limitation, much of the Y chromosome remains unresolved and the discovery rate of this study is lower than expected as we took a very conservative approach to Y chromosome SV identification.

## Discussion

This study extends the comprehensive analyses of large SVs across populations previously published by others[13,22,44]. With 6 samples derived from each of 26 populations comprising the 1KGP, and linked-read sequencing data from 13 samples representing the key populations, our dataset allows us to examine large SVs across the genome and across populations. Genome mapping identified 8.5 times more large insertions than previously reported by the 1KGP from the same samples and 35% more deletions. Not surprisingly, many of the large insertions and deletions were flanked by repetitive elements (such as LINE1) that short-read sequencing cannot sequence across. The bulk of the SVs display an almost identical phylogenetic tree (and very similar population patterns based on PCA) as those derived from SNVs. However, there are 55 complex regions in the genome with variable population patterns and high levels of SV, often consisting of several types of SVs occurring in the same locus (i.e., insertions, deletions, and inversions in different combinations clustered in the same region). The large number of samples from many populations in this study allows us to define the major haplotypes in these regions, and it is clear that the reference genome assembly is but one haplotype of many. Without a comprehensive set of alternate haplotypes representing the variation within and across populations, aligning short-read sequences to the complex regions will lead to errors in analysis.

Based on a saturation analysis[45,46], we estimate the total number of large SVs in the low-complexity regions of human genomes to be around 27,900, and our current dataset has covered around half of the estimate (Supplementary Fig. 25a). We expect to identify 37.7 new large SVs for each additional sample with genome mapping, which is almost twice as many as the 20.9 new large SVs per sample found by sequencing (Supplementary Fig. 25b), demonstrating the power of genome mapping in discovering large SVs.

Multiple alignment of genome maps can help reconstruct a more comprehensive genome assembly and discover new genome patterns. Our study demonstrates how this approach applies to the poorly assembled region at 21p11.2. Notably, large-scale structural features were also spotted at 21p11.2, where individuals carried different types of patterns (B1/B2, D1/D2, and F1/F2) with varying population frequencies. Moreover, two of the genomic regions (B and D) were not previously part of the hg38 assembly. Taken together, genome mapping not only characterizes population patterns of large SV, but also characterizes regions even where the reference is incomplete.

In characterizing CNVs, observed population patterns suggest accelerated evolution in these regions. CNVs play key roles in genome evolution, as they provide raw materials for gene duplication and gene family expansion. They account for phenotypic diversity and variable disease susceptibility. Genome maps facilitate the characterization of CNVs commonly found in hotspots of genomic rearrangement. One variable CNV we characterized affects serum pepsinogen level, a practical predictor of gastric cancer[47,48]. Interestingly, higher copy number in the EAS superpopulation echoes the high prevalence of gastric cancer in this ethnic group[49].

Genome mapping also identified novel genome content not found in the hg38 reference genome sequence. Because these maps are found in multiple samples, there is high confidence that they are real, though absent in the DNA donors of the reference genome. When the new content is sequenced, the non-reference unique insertions can be added to the human genome reference at the correct chromosomal locations, thereby enhancing its usefulness by reducing the fraction of short-read sequencing data discarded because they cannot be mapped back to the human genome reference[16].

Genome mapping data can drive further targeted analyses. For example, one can refine SV breakpoints using linked-reads or other long-read sequencing technologies, although some variants, such as those in highly repetitive regions with repeat units longer than the read length of the sequencing platform, may remain intractable.

Parts of the genome remain inaccessible where the molecules cannot be mapped or assembled because they are devoid of nicking sites or unique nicking site patterns for the nicking enzyme (Nt.BspQI) used. Adding another nicking enzyme (such as Nb.BssSI) might result in unique nicking pattern signatures within the previously inaccessible regions. Additionally, one can use custom CRISPR/CAS9 labeling to target a region that is devoid of nicking sites from any existing commercially available enzyme[41]. Labeling the telomeric repeats with CRISPR-Cas9 nick-labeling, we defined the nicking patterns at the end of the p-arms of acrocentric chromosomes. Future studies will benefit from use of new technologies including the novel Bionano labeling chemistry (Direct Label and Stain) which enables florescent labeling of long molecules with a greater label density than previously available enzymes and without the introduction of systematic fragile site breaks[50]. These improvements allow efficient genome-wide interrogation of variants down to 500 bp.

Our study confirms the abundance of large SVs in the genome and the presence of population-specific SV patterns. It also extends the current genome assembly and identifies additional haplotypes found in complex regions. SV maps from different populations will be very useful in resolving complex regions and provide better references for genome analysis in groups not represented by the DNA donors of the human genome reference sequence assembly. High-speed, cost-effective genome mapping employed in this study makes population-scale genome SV profiling feasible. A major limitation of optical mapping is its lack of sequencing data, such that it does not have single basepair resolution. However, when combined with high-throughput sequencing, genome mapping lays the foundation for haplotype-resolved, structurally accurate medical grade genomes for full genome analysis in the era of precision medicine.

## Methods

**Sample collection**. In all, 156 samples from 26 different populations were studied. From each population, 6 non-related samples (based on pedigree information from the 1KGP), 3 males and 3 females, were chosen (http://www.internationalgenome.org/cell-lines-and-dna-coriell). The corresponding lymphoblastoid cell lines (LCLs) were obtained from the Coriell Cell Repository.

**Data generation and processing**. High-molecular-weight DNA was extracted, nicked, and labeled using the enzyme Nt.BspQI (New England Biolabs (NEB), Ipswich, MA, USA), and imaged using the Bionano Genomics Irys system (San Diego, CA, USA) to generate single-molecule maps for assembly and structural variation analysis.

**Bionano whole-genome mapping**. For high-molecular-weight DNA extraction, cells from the LCLs were washed with phosphate-buffered saline, resuspended in cell resuspension buffer, and embedded into low-melting-point agarose gel plugs (BioRad #170-3592, Hercules, CA, USA). Plugs were incubated with lysis buffer and proteinase K for 4 h at 50 °C. The plugs were then washed, melted, and then

solubilized with GELase (Epicentre, Madison, WI, USA). The purified DNA was subjected to 4 h of drop-dialysis. DNA concentration was determined using Quant-iTdsDNA Assay Kit (Invitrogen/Molecular Probes, Carlsbad, CA, USA), and the quality was assessed based on visual inspection of clarity and viscosity, plus pulsed-field gel electrophoresis.

For DNA labeling, the high-molecular-weight DNA was labeled according to commercial protocols using the IrysPrep Reagent Kit (Bionano Genomics). Specifically, 300 ng of purified genomic DNA was nicked with 7 U nicking endonuclease Nt.BspQI (NEB) at 37 °C for 2 h in NEB Buffer 3. The nicked DNA was labeled with a fluorescent-dUTP nucleotide analog using Taq polymerase (NEB) for 1 h at 72 °C. After labeling, the nicks were repaired with Taq ligase (NEB) in the presence of dNTPs. The backbone of fluorescently labeled DNA was stained with YOYO-1 (Invitrogen).

For data collection, automated electrophoresis of the labeled DNA into the nanochannel array of an IrysChip (Bionano Genomics), followed by automated imaging of the linearized DNA was performed by the Bionano Irys instrument. The DNA backbone (outlined by YOYO-1 staining) and locations of fluorescent labels along each molecule were detected using an in-house image detection software. The set of label locations of each DNA molecule defines an individual single-molecule map.

**10xG linked sequencing**. High-molecular-weight genomic DNA extraction, sample indexing, and partition barcoded libraries were done according to 10X Genomics (Pleasanton, CA, USA), Chromium Genome User Guide and as published previously[15].

**Bionano whole-genome mapping assembly pipeline**. Raw single-molecule maps were de novo assembled into consensus maps using Bionano IrysSolve assembly pipeline with default settings, using pipeline versions 4555 and 4618 (the latter introduced minor bug fixes but no changes to the assembly algorithms).

The assembly pipeline included an implementation of the overlap-layout-consensus algorithm. An initial assembly graph was constructed based on a complete pairwise comparison of single-molecule maps passing a length filter of 150 kb. The graph was then pruned. Spurious edges were removed, and redundant paths collapsed. Draft consensus genome maps representing longest paths from the assembly graph were output. The maps were refined, extended, and merged iteratively in order to generate a set of final consensus maps, which we used as input for hybrid scaffolding with 10xG assemblies and structural variation analysis.

**10xG assembly pipeline**. 10xG raw reads were assembled using the company's Supernova software version 1.1 with default parameters. Output fasta files of the phased supernova assemblies were generated using all possible styles: raw, mega-bubble, pseudohap, and pseudohap2.

**Bionano and 10xG hybrid assembly pipeline**. Pseudohap2 fasta files and Bionano assembled consensus maps were used to generate hybrid assemblies using the Bionano Hybrid Scaffold tool with the following parameters: -N 1 –B 2. To evaluate the quality of the hybrid assemblies, we aligned each hybrid assembly to hg38 using nucmer from the MUMmer package[51] and Assemblytics[52].

**Consensus assembly pipeline**. Assembled consensus maps from each individual were converted to the BNX file format and merged together using Bionano's RefAligner "merge" function (with the following parameters: -merge -bnx -bnxversion 1.2 -randomize). This merged file was then used as input to the Bionano assembly pipeline with default settings to generate the consensus assembly.

**Genome classification**. We classified the genome into distinct categories using the criteria described in Supplementary Fig. 2. First, to determine the genome coverage of the individual assemblies and the meta assembly, we re-aligned these contigs to the reference using RefAligner with the following parameters: -res 2.9 -FP 0.6 -FN 0.06 -sf 0.20 -sd 0.0 -sr 0.01 -extend 1 -outlier 1e-14 -endoutlier 1e-13 -PVendoutlier -deltaX 12 -deltaY 12 -xmapchim 12 5000 -nosplit 0 -biaswt 0 -T 1e-12 -S -1000 -indel -PVres 2 -rres 0.9 -MaxSE 0.5 -MinSF 0.15 -HSDrange 1.0 -outlierBC -outlierLambda 20.0 -outlierType1 0 -xmapUnique 12 -AlignRes 2. -outlierExtend 12 24 -Kmax 12 -resEstimate -M 1 -f -BestRef 0. These parameters output the best single alignment for each area of the assembly while allowing chimeric alignments, especially important for complex regions. We extracted the genome coverage from the resulting alignments.

To define regions of the genome with gaps that affect our analysis, we identified N-gaps in the hg38 reference spanning at least 50 kb, as well as regions in the reference where gaps between in silico predicted BspQI labels spanned at least 100 kb. Since gap sizes in the reference genome are typically estimates, we empirically calculated sizes for a subset of gaps using genome maps (see above), and removed gaps from this set of inaccessible regions if they were able to be sized in at least 25% of samples (Supplementary Data 2).

To build a preliminary list of putative complex regions, we next looked at consensus assembly genome coverage and identified areas covered by two or more meta scaffolds. Genome regions covered by no consensus scaffolds were added to the list if they also had individual assembly coverage of at least 92× (or 46× on chr Y), i.e. at least 60% of samples; these regions were unable to be assembled into a consensus meta scaffold despite high assembly rates in individual samples, suggesting a high rate of structural variation between individuals. Regions with low individual assembly coverage (<92×, or 46× for chr Y) were classified as low coverage. These regions had two primary causes: (1) the region overlapped fragile sites, i.e. areas where BspQI sites were close together on opposite strands and therefore led to frequent breakage of the DNA molecules; and (2) the reference genome differed substantially from individual assemblies, due to either reference assembly errors or very high polymorphism rates, resulting in assembled contigs that did not align to the reference in these areas.

The portion of the genome covered by exactly one consensus scaffold was sorted into two categories depending on its individual assembly coverage. Regions with individual assembly coverage below 92×, or 46× for chr Y, were classified as low coverage, while the remaining regions with high individual assembly coverage were classified as low complexity.

The preliminary list of putative complex regions was manually curated and edited as follows. Each region was visualized in OMView from the OMTools package[53] with the individual assemblies from all 154 samples aligned to the reference with parameters -res 2.9 -FP 0.6 -FN 0.06 -sf 0.20 -sd 0. 0 -sr 0.01 -extend 1 -outlier 1e-14 -endoutlier 1e-13 -PVendoutlier -deltaX 12 -deltaY 12 -xmapchim 12 5000 -nosplit 0 -biaswt 0 -T 1e-12 -S -1000 -PVres 2 -rres 0.9 -MaxSE 0.5 -MinSF 0.15 -HSDrange 1.0 -outlierBC -outlierLambda 20.0 -outlierType1 0 -xmapUnique 12 -AlignRes 2. -outlierExtend 12 24 -Kmax 12 -resEstimate -M 1 -f -BestRef 1. The region was considered complex if it contained at least three different SVs including inversions, translocations, CNVs, and indels, but not including SVs that appeared only once in the dataset. Indels were only considered if they were longer than 10 kb, and regions containing only indels needed to contain at least one that was over 100 kb to be considered complex. For regions that passed the filter, their boundaries were adjusted to capture the complex area and exclude flanking areas with low structural complexity. In cases where parts of the complex regions overlapped with the inaccessible or low-complexity regions (e.g. where a gap or region with low structural complexity was flanked by complex SVs that were merged into a single complex region), those regions were removed from the latter classifications.

Complex regions were annotated with genomic features, ClinVar[17], and OMIM[18] entries with which they overlapped (Supplementary Data 3). Genomic features included SDs[28] (filtered by length ≥ 10 kb, percent identity ≥ 0.95), tandem repeats[28] (filtered by score > 2000), N-gaps, alternate haplotypes in hg38, and telomeric, centromeric, subtelomeric (within 7 Mb of the telomeres), and pericentromeric (within 9 Mb of the centromere) regions. ClinVar entries were included if they were pathogenic/likely pathogenic, not SNVs, and under 10 Mb. OMIM entries were similarly filtered to only include those under 10 Mb.

To determine whether complex regions were overrepresented in different genomic features, we performed 10,000 permutations of the complex regions across the genome using Bedtools[54] shuffle, with the inaccessible regions excluded. For each genomic feature listed above except for N-gaps, we counted the number of overlaps with the permuted complex regions, and determined an empirical $p$ value based on where the real number of overlaps fell in the list of 10,000 permuted values.

**Overview of multiple alignment algorithm**. A detailed description of the multiple alignment algorithm for optical mapping is described elsewhere (Leung et al. (manuscript under review), https://github.com/TF-Chan-Lab/OMTools). Briefly, the multiple alignment analysis pipeline contains three stages: block construction, block sorting, and block merging. The pipeline computes segment matching information from pairwise-alignment results to cluster matching segments into the same group and create a series of collinear blocks. The blocks are then ordered to minimize rearrangement between blocks. Finally, adjacent collinear blocks are merged if their sizes are similar.

**Multiple alignment for selected regions**. Assembled OM contigs were first aligned to a given selected region of interest on the reference hg38 using OMBlast[55]. Next, pairwise alignment of contigs aligning to the selected region was performed; the segment matching information was used to construct the multiple alignment. All multiple alignments employed the same multiple alignment parameters except for analysis in copy number variations where the merging step was skipped. Finally, the multiple alignment was manually refined.

**Structural variation detection pipelines**. We used a modified version of OMSV[19] to identify large (>2 kb) SVs of various types including indels, inversions, duplications, and translocations. We also used Bionano's pipeline for detecting inversions, which were integrated with the OMSV inversions to form a more comprehensive inversion list.

**The modified OMSV pipeline for identifying SVs from optical maps**. The modified version of OMSV identified SVs by comparing the nicking site patterns on the optical maps and the aligned loci of the reference sequence. Three types of comparisons were performed (Supplementary Fig. 4), namely: (1) direct comparison between optical maps of individual DNA molecules with the reference (MR), (2) assembly of the individual optical maps into contigs, followed by comparison of these contigs with the reference (CR), and (3) indirect comparison between the optical maps of individual molecules and the reference by aligning the molecules to the contigs and aligning the contigs to the reference (MCR). The CR and MCR modules were newly added to OMSV in this study. The contigs assembled from individual optical maps spanned longer regions and thus contained more nicking sites for an accurate alignment with the reference. They helped identify large or complex SVs that made direct alignment of optical maps to the reference difficult. The MR module, on the other hand, did not rely on the accuracy of optical map assembly. These several modules were thus complementary to each other. To minimize SV calling errors due to alignment errors, we considered only alignments with a confidence score higher than 9.

**Definition of the in silico reference map**. Both the MR and MCR modules involved the alignment of optical maps to the reference. This reference was produced by in silico digestion, i.e., recording the occurrence locations of all instances of the nicking enzyme motif (5′-GCTCTTC-3′ of NT.BspQI in this study) in the reference genome hg38, including the motif occurrences on both strands.

To match the data resolution of optical maps, all adjacent motif occurrences within 450 bp were merged and represented by a single occurrence at their middle location on the in silico reference map.

**The molecule-reference module**. The MR module in the modified OMSV pipeline was the same as the one in the original OMSV pipeline[19]. Briefly, the optical maps were first aligned to the reference map by combining OMBlast[55] and RefAligner[56] alignments. Then for every pair of adjacent nicking sites on the reference or an optical map, the distances between them on the reference and all optical maps aligned to this locus were compared. Specifically, suppose the distance on the reference was $d_0$, and on the $n$ aligned optical maps, the distances were $d_1$, $d_2$, …, $d_n$, the ratios $d_1/d_0$, $d_2/d_0$, …, $d_n/d_0$ would be used to compute the likelihoods of different hypotheses:

- Null hypothesis $H_0$ that there were no insertions or deletions between the two sites
- $H_{hom}$ that there was a homozygous indel between the two sites
- $H_{het}^{(ins)}$ that there was a heterozygous insertion between the two sites
- $H_{het}^{(del)}$ that there was a heterozygous deletion between the two sites
- $H_{tri}$ that there were two indels between two sites on mating chromosomes, i.e. two different insertions, two different deletions, or one deletion and one insertion

An SV would be called if one of the four alternative hypotheses had a large likelihood ratio to the null, using the default cutoff value of OMSV.

**The contig-reference module**. The CR module compared the contigs assembled from individual optical maps to the reference map. Both the optical map assembly and SV calling were performed using a pipeline by Bionano as described in the corresponding sections. This pipeline did not determine the zygosity of each SV.

**The molecule-contig-reference module**. Since the CR module identified SVs by a direct comparison between contigs and reference without further considering the individual optical maps that supported the SVs, the MCR module supplemented it by using the statistical SV calling method of MR but having optical maps aligned to the reference indirectly through the contigs. Specifically, if nicking site $i$ was aligned to nicking site $j$ on a contig, and $j$ was aligned to nicking site $k$ on the reference using OMBlast, $i$ and $k$ would be aligned indirectly. To ensure the uniqueness of alignment, for each optical map we only considered the contig that the optical map aligned to with the highest assembly score.

**The complex SV detection module**. We also developed a module for identifying three types of complex SVs, namely inversions, translocations, and duplications. Here we use an inversion case to depict the steps of our method in Supplementary Fig. 6.

First, we used the alignments between the contigs and the reference to spot candidate SV regions, and then looked for support of single optical maps from the two-step alignments (MCR) and direct alignments (MR). Specifically, split-alignment allows different parts of a single optical map to be separately aligned to different locations with arbitrary orientations, and the corresponding locations on the reference at which the split alignments start/end are considered breakpoints of candidate SVs:

- Inter-translocations: optical maps are split-aligned to different chromosomes
- Intra-translocations: optical maps are split-aligned to different locations on same chromosomes with distance > 5 Mb
- Inversions: optical maps are split-aligned to the same chromosomes within 5 Mb (Supplementary Fig. 6)
- Duplications: alignments of different segment of the same optical map are (partially) overlapped

Compared with single molecules, the consensus maps are longer and therefore the split alignments of them are more reliable.

For each candidate SV, the molecules supporting the consensus maps with split alignments and the molecules rejecting the SV (supporting the wild type around the breakpoints) were extracted to estimate its score. In both MCR and MR alignments, we selected the molecules around the breakpoints being partially aligned as supporting molecules and the molecules fully aligned with the aligning regions across the breakpoints as rejecting molecules. We define the supporting/rejecting score of a molecule as follows:

$$S_M = \min\left(\frac{n_{m_i}}{n_{m_i} + l_{m_i}}\log\left(n_{m_i}\right), \frac{n_{m_{i+1}}}{n_{m_{i+1}} + l_{m_{i+1}}}\log\left(n_{m_{i+1}}\right)\right), \qquad (1)$$

where $m_i$ and $m_{i+1}$ are the adjacent split maps of molecule $M$, $n_{m_i}$ is the number of mapping sites on the split map $m_i$, and $l_{m_i}$ is the number of extra or missing sites on the split map $m_i$. The rejecting molecules (supporting reference) do not have split maps in the alignment and we presume that they are split at the breakpoints. After scoring, both sets of molecules are de-duplicated by only keeping one record with the largest score for each molecule, and the molecules being considered as both supporting and rejecting are classified to the side with higher scores. The support scores and rejections of all molecules are simply accumulated as total support $S_s$ and total rejection $S_r$. The candidate is discriminated as true SV if $S_s > S_r$.

**Data normalization**. We originally planned to produce optical maps for 156 samples, involving 6 samples from each of the 26 populations. Genome maps produced from two samples (NA21097 and NA21135) subsequently failed to assemble and were discarded, leading to the final list of 154 samples. These 154 samples had good map coverage in general (Supplementary Data 1).

We used two strategies to reduce any bias caused by unequal data coverage in the different samples, namely bootstrap aggregating (bagging) and loose threshold. In the bagging strategy, conceptually for each sample with less than one million aligned optical maps, we repeatedly sampled from the set of aligned genome maps (with replacement) and finally added all of them back such that the sample had around one million aligned optical maps. In our actual implementation, to make the whole procedure more efficient, for each candidate SV we sampled from the molecules aligned to the SV locus only until the number of aligned molecules proportionally corresponded to having one million aligned molecules in total. In the loose threshold strategy, we lowered the minimum number of molecules supporting each SV from 10 (default value) to 6. While both strategies could remove potential bias due to unequal data coverage in the different samples, they could also increase the false positive rate. Comparing the two strategies, the bagging strategy seemed to produce more false positives. Therefore, in the definition of the high-confidence list of SVs, we used only the loose threshold strategy and applied it only to the MCR alignments, which had the highest accuracy among the three sets of alignments. In the full list for explaining the SVs called by 1KGP but missed by our high-confidence list, we applied the bagging strategy to the MR and MCR alignments.

**Inversion detection pipeline**. We also called inversions between the de novo-assembled maps and hg38 using the Bionano Solve 3.1 pipeline. They were obtained by first aligning the assembled maps to the reference using a Multiple Local Alignment algorithm with a maximum likelihood model for evaluating alignments. The resulting alignments were analyzed for inversion signatures.

Inversion breakpoint calls involved neighboring alignments with opposite orientations. Inversions larger than 5 Mb were called as intra-chromosomal translocation breakpoints. To improve detection of small inversions whose inverted regions contain at least four labels, the pipeline searched in a limited space for potential inverted alignments.

The inversions detected by Bionano Solve 3.1 pipeline were combined with the inversions called by the OMSV complex SV detection pipeline to generate a more comprehensive list of inversions.

## Data availability

Genome map and sequencing read data, as well as hybrid assemblies, can be accessed via NCBI under BioProject PRJNA418343 [https://www.ncbi.nlm.nih.gov/bioproject/PRJNA418343].

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

## Acknowledgements

Research reported in this publication was supported by the National Human Genome Research Institute of the National Institutes of Health (NIH) under award R01 HG005946 to P.-Y.K. and M.X. Y.M. was supported by NIH training grant T32 HL007731. A.K.Y.L. and T.-F.C. are partially supported by the HKSAR Research Grants Council General Research Fund 14102014 and Area of Excellence Scheme (AoE/M-403/16). E.Y.C.C. is supported by the Hong Kong PhD Fellowship Scheme. Part of the informatics analysis was run on hardware supported by Drexel's University Research Computing Facility. The content is solely the responsibility of the authors and does not necessarily represent the official views of the National Institutes of Health.

## Author contributions

Conceiving and supervision of study: K.Y.Y., T.-F.C., M.X., and P.-Y.K. Optical map generation: C.C., H.C., A.R.H., C.L., J.M., A.N., J.S., W.-P.W., and E.Y. 10x Genomics Chromium sequencing: A.P. and C.C. Data analysis: M.L.-S., S.P., Y.M., L.L., A.K.Y.L., J. M., E.Y., E.T.L., R.R., N.J., E.Y.C.C., C.Y.L.C., K.H.Y.W., and W.M.

## Additional information

**Competing interests:** E.T.L., A.R.H., A.N., W.-P.W., and H.C. are employees of Bionano Genomics. The remaining authors declare no competing interests.

