## [Peer Review File · Nature Communications]

Reviewer #2 (Remarks to the Author):

The major content and character of this report have not changed since I characterised the first submitted description as hugely impressive, and I remain very positive about the value of the achievements these authors have made in this work.

My last review raised the specific question of the authors' descriptions of tandem repeat (VNTR) regions in this work, and of repeats more generally, where the definitions of the reasons for avoiding or excluding particular classes of region had been rather vague or general. These descriptions have now been updated to become clearer and more explicitly numerical. While it doesn't alter the uncomfortable circumstance that many regions of the genome remain blind spots, it does at least make clear the parameters that define these no-go regions as excluded from study.

Additionally, the Editors asked me to evaluate the extent to which the authors had adequately addressed the concerns expressed by the original reviewer #1, from whom a response was not obtained in the most recent round of review. The main concerns expressed by reviewer #1 in their review of the first revised manuscript NG-A47626R (communicated from Nature Genetics in June 2018) were that the amount of non-reference sequence reported was likely to be overestimated. This reviewer reported that that first revised manuscript "...addressed all my comments and improved the manuscript. ... but the added analyses have also raised concerns about the amount of non-reference sequence found". In the currently submitted version I thought that the analysis of non-reference genomic content and its presentation (lines 309-339) is now thorough and clear in its content and appropriately cautious in its interpretation.

Reviewer #3 (Remarks to the Author):

I remain excited about the implications of this work. The paper has improved and the authors have satisfactorily addressed most of my remaining comments. I appreciate the additional analyses and the clarifications. In cases where we disagree, they have put forward a reasonable argument (although I may not still completely agree).

There are, however, three issues with the revisions that need to be addressed prior to publication:

- 1) In the abstract, they state "We are able to resolve variation in many of the most intractable regions of the genome, including segmental duplications and subtelomeric, pericentromeric, and acrocentric 31 areas." Resolution implies that the regions are solved and that means that the sequence content is known. **An optical map does not provide this.** I suggest change the wording to "map" or "characterize."

2) In regards to the analysis of the acrocentric, my understanding based on their response is that these seven patterns represent unique patterns but in essence are collapses of multiple copies from the acrocentric chromosomes, i.e., "The reviewer is of course correct that the p-arms of acrocentric chromosomes are longer than our contigs and we do not claim that we have this solved." **This observation needs to be made clearer in the text.** If my interpretation is not correct, then please indicate to which acrocentric region these specifically belong to in the figure.

3) I asked previously for a concise plan on how the data generated here could be used to improve the human reference genome. It is unclear, for example, how most geneticists could practically implement the data released as part of this paper. Starting at line 499, the authors have added new text:

"These new content can be added to the human genome reference at the correct chromosomal locations to enhance the usefulness of the human genome reference, reducing the fraction of short-read sequencing data discarded because they cannot be mapped back to the human genome reference."

This does not seem possible. How does the release of optical mapping data for 154 improve mappability of short-read data? Please rework this paragraph.

Response to reviewers' comments:
[Comments in italics, response in blue]

Reviewer #2 (Remarks to the Author):

The major content and character of this report have not changed since I characterised the first submitted description as hugely impressive, and I remain very positive about the value of the achievements these authors have made in this work.

We appreciate the reviewer's enthusiasm for our work.

My last review raised the specific question of the authors' descriptions of tandem repeat (VNTR) regions in this work, and of repeats more generally, where the definitions of the reasons for avoiding or excluding particular classes of region had been rather vague or general. These descriptions have now been updated to become clearer and more explicitly numerical. While it doesn't alter the uncomfortable circumstance that many regions of the genome remain blind spots, it does at least make clear the parameters that define these no-go regions as excluded from study.

Future work with long-read sequencing data will address this issue more fully.

Additionally, the Editors asked me to evaluate the extent to which the authors had adequately addressed the concerns expressed by the original reviewer #1, from whom a response was not obtained in the most recent round of review. The main concerns expressed by reviewer #1 in their review of the first revised manuscript NG-A47626R (communicated from Nature Genetics in June 2018) were that the amount of non-reference sequence reported was likely to be overestimated. This reviewer reported that that first revised manuscript "...addressed all my comments and improved the manuscript. ... but the added analyses have also raised concerns about the amount of non-reference sequence found". In the currently submitted version I thought that the analysis of non-reference genomic content and its presentation (lines 309-339) is now thorough and clear in its content and appropriately cautious in its interpretation.

Appreciate the reviewer's extra effort and his agreement that we addressed reviewer 1's concerns adequately.

Reviewer #3 (Remarks to the Author):

I remain excited about the implications of this work. The paper has improved and the authors have satisfactorily addressed most of my remaining comments. I appreciate the additional analyses and the clarifications. In cases where we disagree, they have put forward a reasonable argument (although I may not still completely agree).

We thank the reviewer for accepting our efforts to improve the paper.

There are, however, three issues with the revisions that need to be addressed prior to publication:

1) *In the abstract, they state "We are able to resolve variation in many of the most intractable regions of the genome, including segmental duplications and subtelomeric, pericentromeric, and acrocentric areas." Resolution implies that the regions are solved and that means that the sequence content is known. **An optical map does not provide this.** I suggest change the wording to "map" or "characterize."*

Agree with the reviewer's point. Changed the wording in the Abstract according to the suggestion.

2) *In regards to the analysis of the acrocentric, my understanding based on their response is that these seven patterns represent unique patterns but in essence are collapses of multiple copies from the acrocentric chromosomes, i.e., “The reviewer is of course correct that the p-arms of acrocentric chromosomes are longer than our contigs and we do not claim that we have this solved.” **This observation needs to be made clearer in the text.** If my interpretation is not correct, then please indicate to which acrocentric region these specifically belong to in the figure.*

Agree with the reviewer. We have clarified this in the revised text by saying, “currently, we cannot localize the acrocentric patterns to specific chromosome arms”. Work is ongoing to place these unique maps to specific acrocentric chromosomes.

3) *I asked previously for a concise plan on how the data generated here could be used to improve the human reference genome. It is unclear, for example, how most geneticists could practically implement the data released as part of this paper. Starting at line 499, the authors have added new text:*

“These new content can be added to the human genome reference at the correct chromosomal locations to enhance the usefulness of the human genome reference, reducing the fraction of short-read sequencing data discarded because they cannot be mapped back to the human genome reference.”

This does not seem possible. How does the release of optical mapping data for 154 improve mappability of short-read data? Please rework this paragraph.

This is a valid point. It is true that the optical maps we produced will not improve the genome reference without sequencing data. In the cases where we do have sequence data, they can be added to the genome reference (e.g., Wong KHY, Levy-Sakin M, and Kwok PY. Nat Commun. 2018 Aug 2;9(1):3040). We have reworked the paragraph as follows:

“When the new content is sequenced, the non-reference unique insertions can be added to the human genome reference at the correct chromosomal locations, thereby enhancing its usefulness by reducing the fraction of short-read sequencing data discarded because they cannot be mapped back to the human genome reference.”